# ChroSegNet: An Attention-Based Model for Chromosome Segmentation with Enhanced Processing

**Xiaoyu Chen** [1] , **Qiang Cai** [2,*] , **Na Ma** [2,*] and **Haisheng Li** [1]

1   School of Computing, Beijing Technology and Business University, Beijing 100048, China
2   School of Biological Science and Medical Engineering, Beijing University of Aeronautics and Astronautics, Beijing 100191, China
*   Correspondence: caiq@th.btbu.edu.cn (Q.C.); manasy@buaa.edu.cn (N.M.)

**Abstract:** In modern medical diagnosis, the karyotype analysis for human chromosome is clinically significant for the diagnosis and treatment of genetic diseases. In such an analysis, it is critically important to segment the banded chromosomes. Chromosome segmentation, however, is technically challenging due to the variable chromosome features, the complex background noise, and the uneven image quality of the chromosome images. Owing to these technical challenges, the existing deep-learning-based algorithms would have severe overfitting problems and are ineffective in the segmentation task. In this paper, we propose a novel chromosome segmentation model with our enhanced chromosome processing, namely ChroSegNet. First, we develop enhanced chromosome processing techniques to realize the quality and quantity enhancement of the chromosome data, leading to the chromosome segmentation dataset for our subsequent network training. Second, we propose our novel chromosome segmentation model "ChroSegNet" based on U-Net. According to the characteristics of chromosome data, we have not only improved the baseline structure but also incorporate the hybrid attention module to ChroSegNet, which can extract the key feature information and location information of chromosome. Finally, we evaluated ChroSegNet on our chromosome segmentation dataset and obtained the MPA of 93.31% and the F1-score of 92.99%. Experimental results show that ChroSegNet not only outperforms the representative segmentation models in chromosome segmentation performance but also has a lightweight model structure. We believe that our proposed ChroSegNet is highly promising in future applications of genetic measurement and diagnosis.

**Keywords:** deep learning; chromosome segmentation; U-Net; attention mechanism; enhanced processing

## 1. Introduction

A healthy human cell has 46 chromosomes, which usually occur in pairs, including 22 pairs of autosomes and one pair of sex chromosomes. Chromosomes are rodlike structures formed by the polymerization of chromatin during mitosis or meiosis. They contain important material required for human genetics and their morphology and structure are closely related to human health. Karyotyping is one of the most important techniques in the field of genetic measurement and diagnosis [1]. Its applications include prenatal screening for chromosomal abnormalities, screening for genetic diseases, etc. Generally, chromosome karyotyping is done on the midterm chromosome micrographs [2], where the number, morphology, and structure of these chromosomes are analyzed and compared by a doctor or specialist, resulting in a karyotype map to help doctors quickly diagnose and predict congenital defects, human genetic diseases, cancers, etc. Therefore, karyotype analysis is of great significance in both research and application. Chromosome karyotyping is divided into three main steps. First of all, the chromosomes are captured and stained with a light microscope. Then, each chromosome was segmented and extracted from the microscopic image of metaphase chromosome. Finally, the extracted chromosomes are classified and

sorted to form a karyotype image with 24 types of chromosomes [3]. For example, Figure 1a shows an image of the 46 types of chromosomes under a 100× microscope and Figure 1b shows the karyotype image of these chromosomes in pairs.

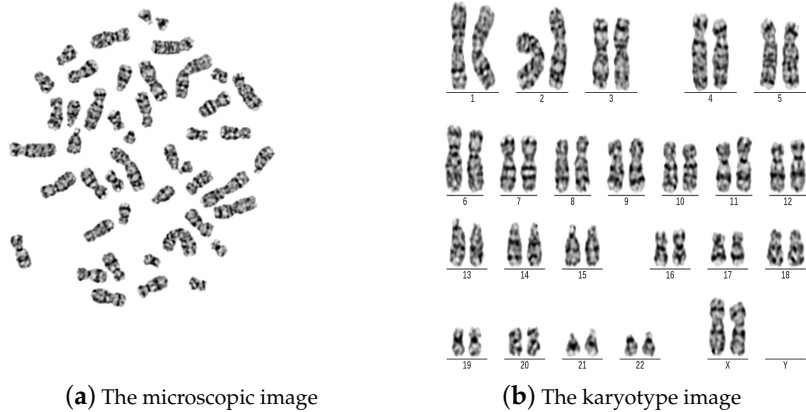

<div align="center">(<b>a</b>) The microscopic image          (<b>b</b>) The karyotype image</div>

**Figure 1.** A chromosome microscopic image (**a**) and the corresponding karyotyping image (**b**).

In early manual karyotyping, doctors would be required to manually extract and classify the chromosomes for analysis, which was not only time-consuming and labor-intensive but also prone to human errors due to its tedious process [4]. Fortunately, the emergence of automatic karyotype analysis system has greatly reduced the workload of karyotype analysis for doctors. It can automatically complete the operations such as chromosome segmentation and classification and finally generate a karyotype map for medical diagnosis. At present, most existing karyotyping systems employ traditional karyotyping methods, which mainly rely on the manual features to complete the chromosome segmentation and classification. For instance, the mainstream traditional chromosome segmentation methods include threshold-based segmentation methods [5], watershed-based segmentation methods [6], fuzzy-clustering-based segmentation methods [7], geometric feature-based segmentation methods [8], etc. The above methods are highly dependent on the manual features and preset parameters thus being limited in their scope of applications. For example, the threshold-based segmentation algorithms rely heavily on the pixel's grayscale information and do not consider the spatial information between pixels. However, due to the strong variability of the chromosome morphological structure and complex distribution, traditional rule-based segmentation methods cannot adapt to such complex situations. In addition, traditional methods still require a lot of manual intervention, which is time-consuming and labor-intensive. Therefore, traditional image segmentation methods still cannot solve the problem of chromosome segmentation well.

With rapid technological advancement, deep learning technology has achieved remarkable results in various fields including computer vision and image processing [9]. Convolutional Neural Network (CNN) is a class of deep neural networks with convolutional structure, which has excellent feature extraction ability, self-learning ability, and low computational cost. In the past decade, CNNs have excelled in various fields of computer vision. Several CNNs including AlexNet [10], Resnet [11], and VggNet [12] have been applied to image classification, image segmentation, and achieved advanced superior performance. In the literature of medical image segmentation, a large number of CNN-based medical image segmentation models have been proposed. Most such models are designed for semantic segmentation [13], e.g., segmenting the cell contours. The task of chromosome image segmentation in this work, however, is to segment (and classify) each chromosome instance. In fact, there is a pressing need to develop the instance-level segmentation models in the field of chromosome segmentation because of its significant applications as discussed earlier.

Specifically, the challenge of chromosome instance segmentation stems from four main points [14]: (1) chromosome data belongs to medical privacy data, which is difficult to obtain, limited in quantity and quality; (2) the chromosomes are prone to distortion, and the same type of chromosomes still have deformation differences; (3) each chromosome image contains a large number of chromosomes, and (4) contacts and overlaps between chromosomes are easy to occur. Because of the above challenges, the existing segmentation networks cannot efficiently complete the task of instance-level chromosome segmentation. In view of the lack of data sources, we worked with the National Engineering Laboratory of Key Technologies for Birth Defect Prevention and Control to obtain chromosome data. However, the quantity and quality of the original chromosome data are far from enough for the experimental standards. Therefore, we designed an enhanced processing pipeline to chromosome data, which assists in our construction of a large-scale chromosome segmentation dataset. To overcome the limitation of the existing segmentation networks, we propose in this paper a novel convolutional neural network for chromosome segmentation, called ChroSegNet. In this network, we designed a new hybrid attention module combining channel attention and spatial attention to extract key feature information and location information of chromosome instances. The channel attention mechanism helps the network to extract key feature information of the target chromosome, while the spatial attention mechanism helps the network to focus on more important spatial information, such as the position relationship between each chromosome. In addition, we carry on the appropriate deepening of the network structure, so that our network can obtain more multidimensional feature information. These improvements would lead to improved feature extraction capability and segmentation accuracy of ChroSegNet, demonstrated in our experimental results reported later.

The main contributions of this paper are summarized as follows.

- We process the chromosome data with particular techniques to realize the quality and quantity enhancement of the chromosome data. Based on the processed data, we constructed our chromosome segmentation dataset containing 13,096 pairs of chromosome data ready for the training of not only our ChroSegNet but also any other CNN models for chromosome processing.
- We propose our end-to-end chromosome segmentation network, i.e., ChroSegNet. ChroSegNet can focus on key feature information and location information of each chromosome through the attention module proposed by us. In addition, the deep-level feature fusion further improves the ability of the network to extract chromosome feature information.

The rest of this paper is organized as follows. In Section 2, we present the related work. In Section 3, we introduce the construction of our enhanced dataset, our enhanced processing, and the structure of ChroSegNet. In Section 4, we evaluate ChroSegNet with 3 evaluation metrics and discuss the experimental results. In Section 5, we conclude our work and discuss the future improvements to our research.

## 2. Related Works

Chromosome segmentation is a branch in the field of medical image segmentation and one of the most critical stages in the process of karyotype analysis. The purpose of chromosome segmentation is to separate the chromosome instances from the complex microscopic chromosome images. Different from other medical images, chromosome microscopic images are susceptible to sensor noises, staining noises, and uneven illumination noises. These noises are due to the irresistible factors in the process of image preparation and acquisition. In addition, chromosomes have the variability of morphological structure and the diversity of contact overlap, which is difficult to identify by traditional methods. The need for hospitals to protect patient privacy has led to difficulties in obtaining chromosome microscopy images; thus, there is a severe lack of data volume. The above problems impose significant challenge in chromosome segmentation. In the early years, many researchers proposed traditional segmentation methods based on specific rules to

segment chromosomes. Ji et al. [15] proposed a rule based on geometric contour analysis to extract chromosomes. Shen [16] and Karvelis [17] proposed segmentation methods based on the watershed algorithm, which are too sensitive to noise thus often leading to oversegmentation. Cao et al. [18] proposed a method based on adaptive fuzzy c-means clustering, which better overcomes the problem of uneven illumination caused by microscope imaging systems and can segment overlapping adjacent chromosomes in different illumination areas. A segmentation method based on spatial variable thresholding was proposed by Grisan et al. [19], which selects the best region for segmentation based on geometric features and pixel distribution. The above traditional segmentation methods, however, have limited performance and are time-consuming and labor-intensive.

In recent years, the excellent performance of deep learning has made it widely utilized in the field of medical images processings. One category of such representative methods is the chromosome segmentation based on convolutional neural network. For example, Esteban et al. [20] proposed an overlapping chromosome segmentation method for MFISH (Multicolor Fluorescence In Situ Hybridization) images, which employs fully convolutional networks [21] (FCN), using spatial and spectral information in an end-to-end manner. Xie et al. [13] proposed a chromosome segmentation model combining Mask-RCNN [22] and geometric correction algorithm. This research achieved the instance segmentation of chromosome microscopic images for the first time. Although this model achieved a high accuracy, its structure is too complicated. When the scale of the real chromosome images is small, the segmentation accuracy of this model drops drastically. Ronneberger et al. [23] proposed U-Net in 2015, which is a network designed based on FCN [21]. Because of the simple structure of the network and the effective use of high and low dimensional feature information, it is suitable for the medical image segmentation with the lack of data and complex image features. Hariyanti et al. [24] proposed a method for semantic segmentation of overlapping chromosomes based on U-Net [23]. This study not only made structural improvements based on the original network such as adding an appropriate number of layers but also used Test Time Augmentation (TTA) to overcome the overfitting problem that occurred during the training process. Compared with previous similar work, the segmentation accuracy is improved, but it is still low due to the lack of improvement for chromosome characteristics. Altinsoy et al. [25] proposed a primitive G-band chromosome image segmentation method based on U-Net. Bai et al. [26] proposed a G-band chromosome segmentation method combining U-Net and YOLOv3. The method consists of two stages. In the first stage, YOLOv3 detects chromosome instances and obtains multiple detection boxes containing one or more chromosome instances. In the second stage, U-Net accurately extracts the single chromosome instance in each detection box. This method has achieved a high segmentation accuracy. However, it still falls into the category of semantic segmentation, and its implementation process is complicated. The huge number of parameters of its dual-network structure lead to a significant increase in the computational cost.

To sum up, the existing deep-learning-based chromosome segmentation methods are still unable give a good balance between scale and accuracy. Based on our chromosome segmentation dataset, we propose ChroSegNet based on lightweight segmentation model, U-Net. U-Net [23] is a fast and accurate network for medical image segmentation, which has been widely applied in various subfields in medical image segmentation [27,28]. For example, it has been applied to segment ultrasound images by various organizations [29–31] and so far has been the best structure for this task [32]. We designed a new attention module according to the characteristics of chromosome instances and incorporated it into U-Net to realize the key information extraction of chromosome instances. In addition, we optimize the network structure on the basis of U-Net [23] to expand the perceptual field, which further improves the segmentation performance.

## 3. Method

In this work, our main focuses are on (1) the construction of a chromosome segmentation dataset and (2) the design of our ChroSegNet model with an effective attention mechanism.

We work with genetic disease laboratory professionals to obtain raw chromosome data. High-quality biomarker slides are prepared by professionals. Then, we use optical microscope camera (including high-resolution camera, optical microscope, and image frame storage board), three-dimensional object loading platform, three-dimensional platform automatic controller, objective lens switching controller, slide glass replacement controller, and computer platform to form a high-level view. The precise three-dimensional optical platform is used to capture and photograph the metaphase chromosomes on the film, and the image data is stored by computer. Again, we used the annotation tool to annotate the 46 chromosomes in each image one by one to obtain label data. Finally, we obtained about 430 RGB microscopic images with a resolution of $1280 \times 1024$ and the corresponding label data in json format.

Since deep learning needs to be driven by large data, and the original image data has problems such as complex noise and insignificant chromosome features, we designed an enhanced processing to convert the limited original data into the expected Enhanced dataset to ensure that the obtained segmentation model has High precision and robustness (see Section 3.1). Based on the Enhanced dataset, we then propose our ChroSegNet for fine chromosome instance segmentation. On the one hand, the hybrid attention mechanism is introduced into ChroSegNet to achieve efficient extraction of chromosome characteristics and location information. On the other hand, we deepen the network structure on the basis of the baseline, in order to obtain more rich multiscale information, which allows the model to be more quickly adapted to microscopic image data with a large number of tiny chromosomes (see Section 3.2).

### 3.1. Enhanced Dataset and Enhanced Processing

The chromosome data we used in this research were obtained from the National Engineering Laboratory of Key Technologies for Birth Defect Prevention and Control. After the steps of chromosome extraction, slide preparation, staining, and digital microscope camera acquisition, the lab's geneticists provided us a total of 430 microscopic images of real G-band chromosomes. Meanwhile, we also obtained the corresponding karyotype images to the chromosomes. All the chromosome images were manually annotated. Subsequently, the original dataset was fed into our enhanced preprocessing module to obtain the chromosome segmentation dataset, as illustrated in Figure 2.

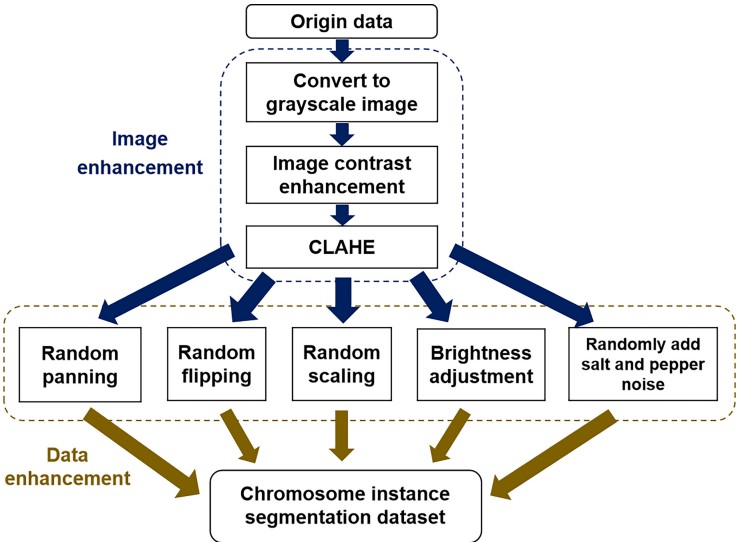

**Figure 2.** Schematic diagram of enhanced processing pipeline.

3.1.1. Image Enhancement Processing

During the chromosome image acquisition, sensor noise, uneven illumination, and various cellular debris can severely degrade the quality of the chromosome images, resulting in a significant impact on the chromosome contours and the banding characteristics. This would affect not only the expert recognition of chromosomes but also the performance of the subsequent chromosome segmentation. The chromosome images provided by geneticists are black and white images since other color features does not help the subsequent chromosome segmentation but increasing the computation. Therefore, we applied gray value adjustment for the initial processing of microscopic images. As there are many impurities and glass slide noises in the chromosome images, we adopted a processing technique combining contrast adjustment and Contrast Limited Adaptive Histogram Equalization (CLAHE) to eliminate the impurities and noises. Meanwhile, the chromosome edges and band features in the image were enhanced. Specifically, the steps of the image enhancement are as follows.

**Step-1 Grayscale Conversion:** Converting all the chromosome microscopic images from RGB to grayscale can reduce the computational dimensionality and the processing time without affecting the image feature extractions. We adopt floating-point arithmetic to replace the R, G, and B channels of original images with the operation results, so as to obtain the grayscale images.

**Step-2 Contrast Stretching:** Due to the unique characteristics of chromosomes such as uneven illumination and blurred imaging, the chromosomes in the original image often lose characteristic information. In addition, irresistible factors, e.g., imaging interference, during the chromosome slide preparation can make chromosome images dark and blurry. Image contrast refers to the difference between the brightest part and the darkest part in an image. In our processing, the contrast stretching operation is performed to map all the pixels in the image to a larger range in the grayscale space. This operation not only effectively reduces the noise interference but also makes the contour and band features of chromosomes more prominent. This operation is formulated in (1):

$$
\begin{cases}
y_g + b > 255 & , y_c = 255 \\
y_g + b \leqslant 255 & , y_c = y_g
\end{cases}
\tag{1}
$$

where $y_g$ represents the grayscale image, $y_c$ represents the contrast-stretched image, and $b$ represents a quantitative value. Since excessive contrast stretching can wash out saturated region of images, we divided the images into two categories: low-light-intensity images and high-light-intensity images. For low-light-intensity images, we set b to 50 for a large stretch. For images with high light intensity, the gray value of the whole image will be higher due to high illumination intensity, so we set b as 20 for fine tuning. Moderate contrast stretching allows further clarification of the image and initial elimination of a large amount of background noise.

**Step-3 Contrast Limited Adaptive Histogram Equalization (CLAHE):** CLAHE was applied after the contrast stretching to ensure further enhancement of the chromosome bands and contouring without amplifying noise. CLAHE is a modified version of adaptive histogram equalization (AHE), which tends to amplify contrast in near-constant areas of the image because of the high concentration of histograms in such areas. This can cause the noise to be amplified in a near-constant region. In CLAHE, the contrast amplification in the vicinity of a given pixel value is given by the slope of the transformation function. This is proportional to the slope of the neighborhood cumulative distribution function (CDF) and therefore to the value of the histogram at that pixel value. CLAHE clipped the histogram to a predetermined value before calculating the CDF. This limits the slope of the CDF and thus the slope of the transform function, both limiting the contrast amplification and therefore reducing the noise amplification problems.

### 3.1.2. Data Augmentation Step

As described in previous section, in this research only 430 pieces of real chromosome data can be obtained, which cannot meet the training requirements of deep-learning-based model. Therefore, to train a model that is more flexible and can better cope with various disturbances, we combine a series of data augmentation algorithms to generate chromosome data and labels in batches for further augmenting the chromosome data. Specifically, the data augmentation techniques employed include random panning, random flipping, brightness adjustment, introducing salt and pepper noise, etc.

After the above processing, an chromosome segmentation dataset in both scale and diversity containing 13,096 pairs of chromosome data was obtained. Each pair of the chromosome data includes a chromosomal microscopic image (JPG file) and a mask label (PNG file) of each chromosome in the corresponding image. Our dataset is divided proportionally, 80% as training set and 20% as testing set.

### 3.2. Network Architecture

ChroSegNet is designed based on U-Net [23] as we reviewed in Section 2. On the one hand, compared with U-Net, in order to take into account the segmentation of small chromosomes, we constructed more subsampling layers and convolutional layers to expand the receptive field. On the other hand, the traditional attention-based U-Net only pays attention to the feature information of a certain dimension (for example, attention gate [33] only pays attention to the spatial dimension.), which is easy to cause the feature information obtained is not comprehensive enough, especially when segmenting images with complex feature information such as electron microscopic images. In contrast, the hybrid attention module we designed focuses on both channel and spatial feature information. The network structure of ChroSegNet is shown in Figure 3.

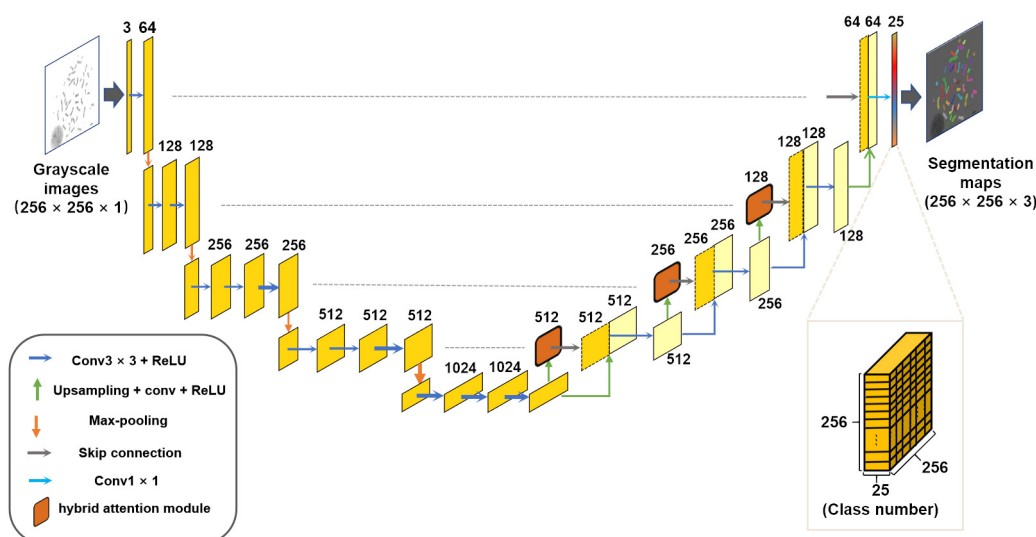

**Figure 3.** ChroSegNet network structure diagram.

ChroSegNet is an encoder–decoder structure. The encoding part is a backbone feature extraction network, which is mainly composed of the convolution layers consisted of $3 \times 3$ convolution kernels, $2 \times 2$ maximum pooling layer, and ReLU activation function. Instead of directly adding attention modules to the original network, we designed more subsampling layers and convolutional layers for ChroSegNet (the newly added parts are shown in bold) to further expand the receptive field and integrate more comprehensive multiscale feature information. The decoding part is an enhanced feature extraction network, which consists of jump connections, upsampling layers, convolution layers, and hybrid attention modules. The hybrid attention modules are incorporated at the end of the skip connections in layers 2, 3, and 4 to generate the multiscale attention information. The attention

information is then input into the feature fusion layer to promote the combination of the high-dimensional key features and the low-dimensional features. This would help the network focuses on more meaningful target regions and suppresses the activation values of the background and the irrelevant regions. The hybrid attention module are mainly responsible for extracting the key parts of the high-dimensional features. However, the first layer of the skip link only contains shallow features and does not involve the information fusion of the high and the low dimensions. Hence, the hybrid attention module is not incorporated in this layer.

The structure of the hybrid attention module is shown in Figure 4, where $S$ represents the feature map of the current jump-connected input, $X$ represents the feature map of the current input, $X'$ represents the output value of the channel attention module, $X''$ represents the output of the hybrid attention module, $\alpha$ represents the attention coefficient, and $\beta$ represents the spatial attention coefficient.

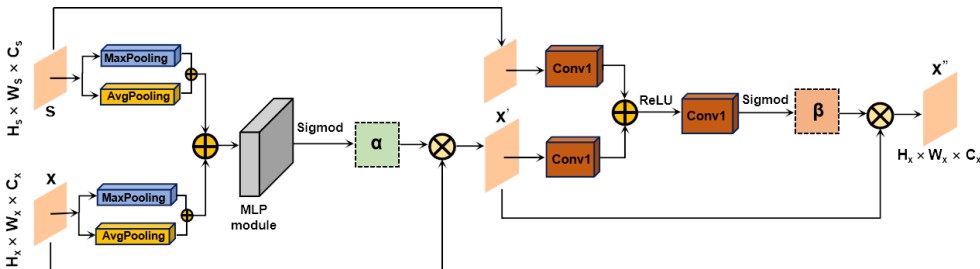

**Figure 4.** Diagram of the hybrid attention module.

The feature map of the encoder input through the jump connection layer and the current input feature map of the decoder first enter the channel attention module to extract the channel attention information. The process of the channel attention module is as follows: the average pooling layer and the maximum pooling layer are used to compress the spatial dimension of the input feature mapping. Among them, the average pooling layer is a commonly used means of spatial information aggregation, and the maximum pooling layer has been proved to be able to collect the key clues of the features of different objects [34]. Therefore, we adopt the combination of the two pooling methods to obtain more representative information. Then, the descriptor information after secondary fusion is input into Multilayer Perceptron (MLP) including convolution layer and ReLU layer to obtain channel attention mapping. Finally, Sigmod is used to change it into the channel attention coefficient ($\alpha$) between 0 and 1 and multiply with the current input feature map to obtain the feature map with the channel attention information ($X'$). Next, the feature map with channel attention information ($X'$) and the encoder feature map are input into the spatial attention module. The process of the spatial attention module is as follows: The $1 \times 1$ convolution layer is used to compress the channel dimension of the input feature map, and then the activation function ReLU and the $1 \times 1$ convolution layer are used to obtain the spatial attention map. Finally, Sigmod is used to obtain the spatial attention coefficient ($\beta$) and multiply with the current input feature map to obtain the feature map with spatial attention information ($X''$). In conclusion, the hybrid attention module that we designed can focus on both "what" and "where" questions of chromosome instance.

## 4. Experiments and Results

### 4.1. The Experimental Details

#### 4.1.1. The Experiment Setup

The experiments were run on the computer platform with AMD Ryzen 7 3700X processor and NVIDIA RTX3090 GPU. The operating system used is Windows 10. The network is coded with PyTorch 1.6.0, and the enhanced processing part of the chromosome dataset is implemented based on opencv-python 4.5.2. First, the proposed image enhancement is performed on all the original images. Then, data augmentation is applied to the enhanced

images to obtain our chromosome segmentation dataset containing 13,096 pairs of chromosome data. Next, 10,477 pairs of data were randomly selected from the dataset at the ratio of 80% and 20% as the training set, and 2619 pairs of data were used as the testing set. Table 1 lists the training and testing hyperparameter settings, where Epoch is set to 100 times and Batch size is set to 4 images per batch. The gradient descent optimizer selected is Adam. Since this model is trained from scratch, the learning rate is adjusted to 0.000001 to ensure better convergence. We used our computer platform to train iteration of 100 epochs on ChroSegNet, which took 36 h in total. In addition, in order to test the implementation time, we used ChroSegNet to segment 30 chromosome microscopic images, and the operation time was 48 seconds, with an average of 1.60 s per image.

**Table 1.** Hyperparameter setting.

| Hyperparameters | Details |
| :---: | :---: |
| Epoch | 100 |
| Batch size | 4 |
| Optimizer | Adam |
| Learning rate | 0.00001 |
| Threads number | 4 |

4.1.2. Loss Function

Cross-entropy can enable the network to learn the similarity between the predicted value and the ground truth more quickly. Specifically, compared with loss functions such as Mean Square Error (MSE), the gradient updating amplitude is faster, and the problem of gradient dispersion will not occur. Meanwhile, since chromosome instance segmentation involves multiclassification, we use multivariate cross-entropy as the loss function of the network instead of binary cross-entropy. The multivariate cross-entropy loss function is calculated as follows (2):

$$CEloss(x, class) = -log\left(\frac{exp(x[class])}{\sum_j exp(x[j])}\right) \tag{2}$$

where $x$ represents the output of the last layer of the network, *class* represents the label of a chromosome class to be calculated, and $x[j]$ represents the predicted value of the output of all chromosome classes. Specifically, softmax converts the final output value of the network into a value between 0 and 1 (the sum of the predicted probabilities of all the categories is 1). Then, it extracts the corresponding value according to the label index to calculate the final losses.

*4.2. Enhancement Processing*

As described in Section 3.1, the original chromosome images (430 in total) were processed for quality and quantity enhancement. A sample chromosome image is shown in Figure 5a. First, we convert the original RGB image to a grayscale image with a size of 516 × 516, as shown in Figure 5b. It can be observed that the image well remains its structural information after the conversion. Second, the image contrast stretching operation is performed to stretch the image contrast to the range of [0, 255]. As shown in Figure 5c, we can see that the contours of the chromosomes in the image become clearer, and some impurities in the background are eliminated or attenuated. Finally, CLAHE is performed to further enhance chromosome structural features without amplifying background noise, especially the strip feature. Our proposed method includes the above steps, and the final result is shown shown in Figure 5d. The originally blurred band features in the chromosome image become clearer and more distinguishable, with distinct outline features for each chromosome instance. At the same time, the background noise is well suppressed.

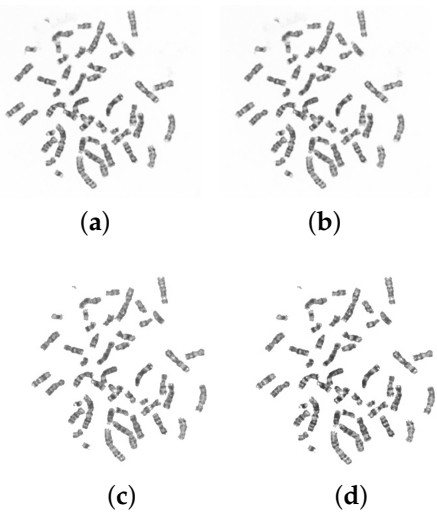

**Figure 5.** Schematic diagram of image enhancement processing results. (**a**) Original; (**b**) grayscale; (**c**) contrast stretched; (**d**) proposed method.

The data after image enhancement is sent to our proposed data augmentation step, which mix the various methods including random translation, random brightness adjustment, random scaling, etc. to process the input data and generate new generated data. In the first stage, the data augmentation is carried out on 430 pieces of data, and 30 pieces of generated data are generated for each pieces of data. At this time, a total of 12,900 new data are generated. In the second stage, we manually screened the new data generated, removed the generated data with low quality (excessive noise generation, excessive translation, etc.), and finally obtained 12,666 generated data. The generated data and the original data constitute the chromosome segmentation dataset containing 13,096 pieces of data. The generated data includes generating images and corresponding label files.The new sample obtained by data augmentation is shown in Figure 6.

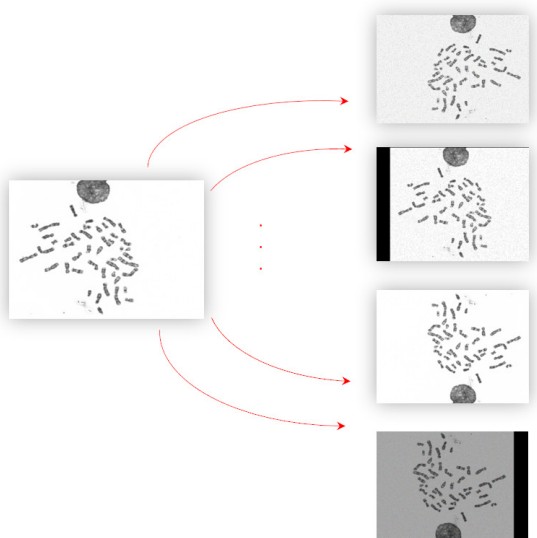

**Figure 6.** Schematic diagram of data augmentation to generate new samples.

*4.3. Evaluation Metrics and Chromosome Segmentation Results*

4.3.1. Evaluation Metrics

In this research, we evaluate the performance of the chromosome segmentation with three metrics: IoU, PA, and F1-score. IoU, which stands for Intersection over Union, represents the ratio of the intersection and union of the prediction result of a certain

category and the ground truth. In other words, it is the ratio of the intersection and union between the predicted mask and the real mask. PA stands for Pixel Accuracy, which reflects the ratio of the number of pixels that correctly predicted to belong to a chromosome category to the total number of pixels in the image. F1-score is the harmonic mean of accuracy and recall. It combines the results of accuracy and recall and can better measure the comprehensive performance of a model.

### 4.3.2. Chromosome Segmentation Results

Since there are no publicly available chromosome datasets, we trained our proposed model on the chromosome segmentation dataset we constructed. As there is no publicly available code on chromosome segmentation, we compare the chromosome segmentation performance of ChroSegNet with representative segmentation models in recent years to demonstrate its effectiveness. Furthermore, we conducted separate ablation experiments to verify the effectiveness of our enhanced processing and model improvements. Table 2 shows the segmentation results by ChroSegNet for 24 categories of chromosome instances based on the above three metrics. Classes 1–22 denote chromosome classes 1–22, and classes x and y correspond to the two sex chromosomes, respectively. It can be observed from the table that ChroSegNet achieves the best IoU and F1-score on class 1, reaching 90.17% and 94.83%, respectively. As for class 2, our model achieves the best PA reaching 95.29%. The segmentation results for smaller chromosomes, e.g., class 22 and class Y, are not that high but the lowest IoU score can still reach more than 80%, and the lowest F1-score can reach more than 88%. The results show that ChroSegNet can achieve high precision and robust segmentation for each category of chromosome.

**Table 2.** Segmentation performance of ChroSegNet for various chromosome instances.

| Class (No.) | IoU (%) | PA (%) | F1 (%) |
|:---:|:---:|:---:|:---:|
| 1 | **90.17** | 95.23 | **94.83** |
| 2 | 90.04 | **95.29** | 94.76 |
| 3 | 89.90 | 95.26 | 94.68 |
| 4 | 89.27 | 94.87 | 94.33 |
| 5 | 89.11 | 94.64 | 94.24 |
| 6 | 89.40 | 95.01 | 94.40 |
| 7 | 88.23 | 94.05 | 93.75 |
| 8 | 87.86 | 93.98 | 93.54 |
| 9 | 87.26 | 93.61 | 93.19 |
| 10 | 88.32 | 94.46 | 93.80 |
| 11 | 88.19 | 94.28 | 93.72 |
| 12 | 88.23 | 94.44 | 93.75 |
| 13 | 86.72 | 93.20 | 92.89 |
| 14 | 85.55 | 92.61 | 92.21 |
| 15 | 86.37 | 92.42 | 92.69 |
| 16 | 84.85 | 92.70 | 91.80 |
| 17 | 85.21 | 92.03 | 92.01 |
| 18 | 85.15 | 91.91 | 91.98 |
| 19 | 81.64 | 90.81 | 89.89 |
| 20 | 82.90 | 91.22 | 90.65 |
| 21 | 80.67 | 88.18 | 88.68 |
| 22 | 80.42 | 88.23 | 89.15 |
| X | 88.47 | 94.23 | 93.88 |
| Y | 82.33 | 90.54 | 90.31 |

Bold represents the best performance value for each class.

We compare and analyze ChroSegNet with other segmentation models in terms of segmentation performance and model parameters shown in Table 3.

**Table 3.** Segmentation performance analysis of different segmentation models with ChroSegNet.

| Models | MIoU (%) | MPA (%) | F1 (%) |
|---|---|---|---|
| Mask-RCNN | 79.91 | 83.49 | 85.27 |
| PSP-Net | 69.84 | 75.02 | 74.77 |
| DeepLabV3 | 64.95 | 69.55 | 71.34 |
| ChroSegNet | **86.97** | **93.31** | **92.99** |

Bold represents the best performance value for each model.

Obviously, ChroSegNet achieves the best performance on every evaluation metric while the DeepLabV3's performance is the lowest. We think that the poor performance of DeepLabV3 is due to its hollow convolution structure and noncontinuous convolution kernel. In fact, the microscopy images of chromosome contain a large number of small chromosomes, and the DeepLabV3's discontinuous convolutional structure can easily lead to the loss of small target features. Although the segmentation performance of Mask-RCNN is better than that of DeepLabV3 and PSP-Net, it still significantly underperforms compared to ChroSegNet. We believe this is caused by the large size of Mask-RCNN, which cannot be fully trained on small-scale datasets due to its complex structure and huge number of parameters. In summary, although the above benchmarking networks can have good segmentation performance in their respective fields, they still cannot achieve high-precision instance segmentation for chromosomes. In contrast, we can see that ChroSegNet shows not only a significantly higher segmentation accuracy as 93.31% but also an F1 value as high as 92.99%, which indicates that it can well balance segmentation accuracy and recall rate. This indicates that ChroSegNet is more suitable for chromosome segmentation than other models. Table 4 shows the number of parameters for each segmentation model, which reflect the computing resources that a model needs to occupy. It can be seen from Table 4 that Mask-RCNN requires the most computing resources. PSP-Net has the fewest parameters due to its simple structure. Although ChroSegNet adopts hybrid attention modules and further deepens the network structure, the number of parameters of ChroSegNet remains at a low level, i.e., only 2.89 M more parameters than PSP-Net. This data shows that our model is more lightweight and can be considered for practical application in the future.

**Table 4.** Comparison of parameters of different models and ChroSegNet.

| Models | Total Params |
|---|---|
| Mask-RCNN | 65.121 M |
| PSP-Net | 46.718 M |
| DeepLabV3 | 54.715 M |
| ChroSegNet | 49.608 M |

Table 5 shows the effect of our enhanced processing by comparing segmentation results of U-Net on the original dataset and our chromosome segmentation dataset. It can be observed from the table that the final result of U-Net training on the chromosome segmentation dataset has a huge improvement compared with the results obtained on the original dataset. Specifically, the MIoU has increased by 11.35%, the MPA has increased by 7.89%, and the F1-score has increased by 7.80%. This proves that our enhanced processing effectively improves the quality, quantity, and the diversity of the original chromosome dataset, which can promote the model to learn the target features more efficiently. Due to the lack of publicly available large-scale datasets in the field of chromosome segmentation, the chromosome segmentation dataset that we constructed lays a foundation for future research on chromosome segmentation or relevant processing.

**Table 5.** Enhanced processing effect comparison.

| Dataset | MIoU (%) | MPA (%) | F1 (%) |
|---------|----------|---------|--------|
| Original | 66.79 | 79.25 | 79.78 |
| Enhanced | **78.14** | **87.14** | **87.58** |

Bold represents the best performance value for each Dateset.

Table 6 shows the performance comparison of U-Net, U-Net+ATT (U-Net with hybrid attention module), U-Net* (U-Net with improved backbone network), and ChroSegNet on our enhanced dataset mentioned above (all three indicators are the average of the corresponding metrics for all categories). It can be observed that the MIoU, MPA, and F1 of the two improved baseline models (U-Net+ATT and U-Net*) are significantly improved, especially U-Net with hybrid attention module. ChroSegNet achieved the best results by combining the two improvements, which confirms the effectiveness of the combination of the two improvements.

**Table 6.** Segmentation performance of different modules based on the baseline model (U-Net).

| Models | MIoU (%) | MPA (%) | F1 (%) |
|--------|----------|---------|--------|
| U-Net | 78.14 | 82.14 | 80.09 |
| U-Net+ATT | 85.50 | 92.60 | 92.13 |
| U-Net* | 80.73 | 85.56 | 83.07 |
| ChroSegNet | **86.97** | **93.31** | **92.99** |

Bold represents the best performance value for each model.

Figure 7 shows the segmentation mask comparison between ChroSegNet and U-Net on multiple chromosomal microscope images. It can be observed from the comparison of Group A that U-Net still segmented part of the background as chromosome instances by mistake. In contrast, no background was misclassified as chromosome instances by ChroSegNet.

In the Group B comparison, a large number of single chromosomes are divided into multiple chromosomes by mistakes in the segmentation mask of U-Net, and some adjacent chromosomes are not effectively segmented. On the contrary, these errors rarely occur in ChroSegNet's mask because ChroSegNet can obtain more critical chromosomal characteristics and location information. In Group C, due to the denser chromosomes, U-Net has yielded more incorrect segmentation masks, including wrongly classifying the entire chromosome or part of the chromosome, and unsegmenting overlapping chromosomes. On the contrary, the segmentation mask of ChroSegNet is more accurate, which benefits from its deep structure and attention information acquisition. In summary, compared with the original U-Net model, ChroSegNet has higher chromosome segmentation accuracy and better robustness.

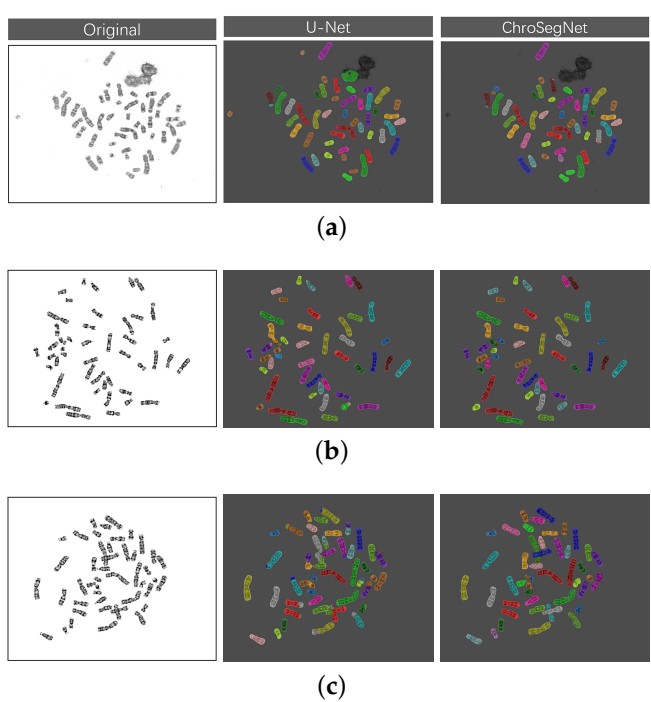

**Figure 7.** Comparison of segmentation masks between ChroSegNet and U-Net on chromosome microscopy images. From left to right are the original image, the U-Net segmentation mask, and the ChroSegNet segmentation mask. (**a**) Group A comparison; (**b**) Group B comparison; (**c**) Group C comparison.

## 5. Conclusions

In this paper, we design our ChroSegNet model with an effective attention mechanism for accurate chromosome segmentation. On the basis of considering the lightweight model, our ChroSegNet not only has a deep network structure but also has the hybrid attention structure, which is responsible for simultaneously extracting key features and position information of chromosomes. Experimental results show that ChroSegNet is more suitable than most CNN models to deal with chromosomes with complex structure and changeable position. To construct a dataset for our modeling training, we cooperated with the laboratory to acquire chromosome data and proposed enhanced processing to enhance the quality and quantity of chromosome data, resulting in the chromosome segmentation dataset which with large scale and high quality. Our experimental results show that the performance of the segmentation model trained with our dataset is better than that trained with the original dataset.

However, the current ChroSegNet is still limited in the following aspects. On the one hand, the segmentation performance of ChroSegNet is relatively limited for the same class of chromosomes with large deformation. For this limitation, we plan to design and incorporate additional branches in ChroSegNet to learn and utilize the chromosome shape information for improved segmentation accuracy in our future work. On the other hand, the segmentation performance of ChroSegNet for overlapping chromosomes still needs to be improved. We plan to further optimize the network structure in future work, such as attempting to model ROI as multiple layers and detecting overlapping chromosomes separately during segmentation.

**Author Contributions:** Writing—original draft, X.C.; writing—review and editing, X.C. and Q.C.; software, X.C.; validation, Q.C., N.M. and H.L.; conceptualization, X.C., Q.C. and N.M.; methodology, Q.C. and N.M.; formal analysis, Q.C. and H.L.; resources, N.M.; supervision, H.L. All authors have read and agreed to the published version of the manuscript.

**Funding:** This research received no external funding.

**Institutional Review Board Statement:** Institutional Review Board Statement: Approval document for clinical scientific research projects of the Medical Ethics Committee of the Chinese People's Liberation Army General Hospital. approval number: S2018-132-01.

**Informed Consent Statement:** Informed consent was obtained from all subjects involved in the study. Written informed consent has been obtained from the patient(s) to publish this paper.

**Data Availability Statement:** Data available on request due to restrictions eg privacy or ethical The data presented in this study are available on request from the corresponding author. The data are not publicly available due to the data involves patient chromosome data.

**Conflicts of Interest:** The authors declare no conflict of interest.

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
