# Peer review of "ChroSegNet: An Attention-Based Model for Chromosome Segmentation with Enhanced Processing"

_applsci, doi:10.3390/app13042308_

Round 1

Reviewer 1 Report

The grayscale conversion formula is standard and seemed to be redundant to be included.

in equation 2, how b was determined? setting everything above 255 to 255 may cause washed out images in saturated region. How would that impact image quality.

section 3.1.2, how would images underwent different data enhancement be distributed. Are there even number of each category?

Is the unique feature for ChroSegnet having more subsampling and convolution layers, apart from conventional attention based U-net? I feel the description of network architecture could be improved/condensed to more clearly state its new features.

I feel there is little need to include equation 4, 5, 6, because these are standard metrics.
